# ACE-2, TMPRSS2, and Neuropilin-1 Receptor Expression on Human Brain Astrocytes and Pericytes and SARS-CoV-2 Infection Kinetics

**DOI:** 10.3390/ijms24108622

**Published:** 2023-05-11

**Authors:** Johid Reza Malik, Arpan Acharya, Sean N. Avedissian, Siddappa N. Byrareddy, Courtney V. Fletcher, Anthony T. Podany, Shetty Ravi Dyavar

**Affiliations:** 1Antiviral Pharmacology Laboratory, Department of Pharmacy Practice and Science, College of Pharmacy, University of Nebraska Medical Center, Omaha, NE 68198, USA; jomalik@unmc.edu (J.R.M.); sean.avedissian@unmc.edu (S.N.A.); cfletcher@unmc.edu (C.V.F.); 2Department of Pharmacology and Experimental Neuroscience, University of Nebraska Medical Center, Omaha, NE 68198, USA; arpan.acharya@unmc.edu (A.A.); sid.byrareddy@unmc.edu (S.N.B.); 3Department of Genetics, Cell Biology, and Anatomy, University of Nebraska Medical Center, Omaha, NE 68198, USA; 4Department of Biochemistry and Molecular Biology, University of Nebraska Medical Center, Omaha, NE 68198, USA; 5Division of Clinical Microbiology, Department of Laboratory Medicine, Karolinska Institute, 17177 Stockholm, Sweden

**Keywords:** ACE-2, TMPRSS-2, Neuropilin-1, SARS-CoV-2, COVID-19, astrocytes, pericytes, brain microvascular endothelial cells and blood-brain-barrier

## Abstract

Angiotensin Converting Enzyme 2 (ACE-2), Transmembrane Serine Protease 2 (TMPRSS-2) and Neuropilin-1 cellular receptors support the entry of SARS-CoV-2 into susceptible human target cells and are characterized at the molecular level. Some evidence on the expression of entry receptors at mRNA and protein levels in brain cells is available, but co-expression of these receptors and confirmatory evidence on brain cells is lacking. SARS-CoV-2 infects some brain cell types, but infection susceptibility, multiple entry receptor density, and infection kinetics are rarely reported in specific brain cell types. Highly sensitive Taqman ddPCR, flow-cytometry and immunocytochemistry assays were used to quantitate the expression of ACE-2, TMPRSS-2 and Neuropilin-1 at mRNA and protein levels on human brain-extracted pericytes and astrocytes, which are an integral part of the Blood-Brain-Barrier (BBB). Astrocytes showed moderate ACE-2 (15.9 ± 1.3%, Mean ± SD, n = 2) and TMPRSS-2 (17.6%) positive cells, and in contrast show high Neuropilin-1 (56.4 ± 39.8%, n = 4) protein expression. Whereas pericytes showed variable ACE-2 (23.1 ± 20.7%, n = 2), Neuropilin-1 (30.3 ± 7.5%, n = 4) protein expression and higher TMPRSS-2 mRNA (667.2 ± 232.3, n = 3) expression. Co-expression of multiple entry receptors on astrocytes and pericytes allows entry of SARS-CoV-2 and progression of infection. Astrocytes showed roughly four-fold more virus in culture supernatants than pericytes. SARS-CoV-2 cellular entry receptor expression and “in vitro” viral kinetics in astrocytes and pericytes may improve our understanding of viral infection “in vivo”. In addition, this study may facilitate the development of novel strategies to counter the effects of SARS-CoV-2 and inhibit viral infection in brain tissues to prevent the spread and interference in neuronal functions.

## 1. Introduction

Coronavirus disease 2019 (COVID-19) caused by severe acute respiratory syndrome coronavirus-2 (SARS-CoV-2) has emerged as a major pandemic impacting lives worldwide. Johns Hopkins University reported 676.6 million COVID-19 cases and over 6.88 million total deaths as of 10 March 2023, the last date of data collection [1]. Recent evidence suggests that SARS-CoV-2 infects epithelial cells localized to lung and gut tissues and additionally some brain cells [2,3,4]. Neurological abnormalities with symptoms such as confusion and disorientation are observed in some SARS-CoV-2 infected individuals [5,6]. The severity of neurological diseases, including meningitis and encephalitis, increased in COVID-19 patients with convulsions [7]. A mechanistic understanding of neurological abnormalities during SARS-CoV-2 infection in COVID-19 patients was obtained from a meta-analysis of 153 patients who exhibited neuro-physiological disorders and later advanced to psychosis [8]. FDG-Positron-Emission tomography/computer tomography of frontal lobe and cerebellum regions following SARS-CoV-2 infection in COVID-19 patients showed acceleration of encephalitis due to lowering of metabolism [9].

Inflammation is a major driver of aberrant lung function and progressive SARS-CoV-2 infection and is frequently associated with abnormal brain function in COVID-19 patients [10]. More aggressive immune responses may cause damage to brain cells in COVID-19 patients and may cause Guillen-Barré syndrome (GBS), a disorder of the peripheral nerves, and even Miller–Fisher syndrome, a rare nerve disease [11,12,13,14,15]. Higher levels of secretory cytokines that regulate inflammation and autoantibodies are associated with central nervous system (CNS) complications in COVID-19 patients [16,17,18].

Effects of SARS-CoV-2 are not only associated with the respiratory system but also commonly associated with the gastrointestinal system [19], cardiovascular system [20], reproductive system [21,22], and neurological system [23,24]. Brain infection with SARS-CoV-2 has been associated with neurological disorders and cerebral artery stroke in COVID-19 patients [18,24,25]. SARS-CoV-2 was detected in the cerebrospinal fluid (CSF) of a 34-month-old child with encephalitis and in the brain tissue of an adult patient [23,26,27]. However, the effects of SARS-CoV-2 infection on the blood-brain barrier (BBB) and cell types located in this layer were rarely investigated.

Entry of coronaviruses into the brain compartment occurs through BBB transport passing the cerebral microvascular endothelial monolayer barrier [28,29,30]. Invasive entry of SARS-CoV-2 into brain tissue requires the contribution of molecular receptors on brain cells such as ACE-2 (Angiotensin Converting Enzyme 2) and Neuropilin-1 [31]. Transmembrane Protease 2 (TMPRSS-2) is a cellular protease that primes viral spike protein to attach to ACE-2 and facilitates SARS-CoV-2 entry into target cells [32]. Eminence capillaries and tanycytes of the hypothalamus express TMPRSS-2 and ACE-2 and facilitate SARS-CoV-2 infection of brain tissue [33]. The role of host cofactors or tight junctional proteins involved in SARS CoV-2 transport through the BBB remains unidentified. Human brain microvascular endothelial cells (hBMVEC) express low levels of ACE-2 and the recombinant SARS-CoV-2 spike protein enhances ACE-2 expression [34]. In contrast, human cortical astrocytes express ACE-2 and Neuropilin-1 [35,36]. However, Neuropilin-1 association with neuronal pericytes, which are an integral part of BBB, remains unknown. 

In this study, we analyzed the expression of ACE-2, Neuropilin-1, and TMPRSS-2 both on the surface of astrocytes and pericytes and at the mRNA level, using highly specific Taqman ddPCR, flow-cytometry, and immunoassays. Our data provide evidence on ACE-2, TMPRSS-2 and Neuropilin-1 expression on primary pericytes and astrocytes and the potential involvement of these proteins in SARS-CoV-2 infection of these cell types.

## 2. Materials and Methods

**Cells and Culture system**—Primary human brain extracted astrocytes (CAT# 1800), pericytes (CAT# 1200) and human brain microvascular endothelial cells (hBMVECs, CAT# 1000) as well as human lymph node extracted endothelial cells (hLNECs, CAT# 2500) were purchased from ScienCell Research Laboratories (SCRL), Carlsbad, CA, USA. Cell culture media and the growth supplements required for culturing brain cells including astrocyte media (AMCAT, CAT# 1801) with growth supplement (AGS, CAT# 1852), pericyte media (PMCAT# 1201) with growth supplement (PGS, CAT# 1252) were purchased from the manufacturer, SCRL, Carlsbad, CA, USA. hBMVECs and hLNECs were cultured in an endothelial cell medium (ECM, CAT# 1001) with a growth supplement (ECGS, CAT# 1052) that was purchased from SCRL, Carlsbad, CA, USA. Fetal bovine serum (FBS, CAT# 0010) and penicillin/streptomycin solution (P/S) (CAT# 0503) media supplements were purchased from SCRL, Carlsbad, CA, USA. The cell thawing procedure was followed as described by the manufacturer. Per the experimental requirement, cells were grown in 25/75/150 cm^2^ tissue culture flasks (TPP # 90076). Tissue culture flasks and plates used for culturing brain cells and hLNECs were pre-coated with bovine fibronectin (2 µg/mL) (SCRL, CAT# 8248). Confluent cell monolayers were passaged according to the manufacturer’s instructions. In this, cells were washed with DPBS (Dulbeccos, CAT# 1960454) and detached from the surface via treatment with 0.25% trypsin (CAT# CC-5012, Lonza, USA) for 1–2 min. Following trypsin treatment, cells were mixed with FBS followed by adding a culture medium. Cells were pelleted by centrifugation at 1000 rpm for 5 min at room temperature (RT or 25 °C). Cell count was performed by mixing the cell suspension with trypan blue at a 1:1 ratio and 10 µL of suspension was loaded onto the slide provided by the manufacturer (Invitrogen, Eugene, OR, USA) and counted in Countess cell counter (Invitrogen, Chicago, IL, USA). Cells below 5 passages were used for the experiments to maintain the integrity of the phenotype of cells.

**VERO-E6 and Jurkat cell lines**—VERO and Jurkat cells were purchased from ATCC, Manassas, VA, USA. VERO cell line was cultured in DMEM (ATCC 30-2002), and Jurkat cells were cultured in RPMI media, and both were supplemented with 10% FBS and P/S.

**Flow cytometry assays**—ACE-2, TMPRSS-2 and Neuropilin-1 expression on brain cells was analyzed by flow-cytometry using anti-ACE-2 Alexa Fluor-647 conjugated polyclonal antibody (Ab) (CAT# FAB933R, R&D, Minneapolis, MN, USA), Alexa Fluor-488 conjugated anti-TMPRSS-2 Ab (Santacruz Biotechnology, Dallas, TX, USA, CAT#SC-515727) and PE-conjugated anti-Neuropilin-1 Ab (CAT#FAB3870P-100, R&D, Minneapolis, MN, USA) respectively. Zombie aqua (CAT# 423101) dye (Biolegend, San Diego, CA, USA) was used to identify live and dead cells. Approximately 0.7 million brain cells were washed with DPBS and stained with Zombie aqua at a 1:1000 dilution for 15 min at RT in the dark. All staining steps were performed at RT. After a PBS wash, cells were incubated in FACS buffer (5% FBS in DPBS) for 10 min. Cells were rewashed with DPBS and incubated with Alexa Fluor-647 conjugated ACE-2 Ab (6 µg/mL) for 30 min. in the dark. PE-conjugated anti-Neuropilin-1 Ab (2 µg/mL) and Alexa Fluor 488 conjugated anti-TMPRSS-2 Ab (Santacruz Biotechnology, Dallas, TX, USA, CAT#SC-515727) (2 µg/mL) or as a cocktail of all three antibodies. After completion of the staining procedure, cells were washed with DPBS and were fixed with 100 µL 4% paraformaldehyde (Thermo Scientific, Waltham, MA, USA, CAT# J19943-K2). Fixed cells were washed with DPBS and analyzed for expression in a BD FACSDiva 8.0.2 (USA). Due to technical difficulties, TMPRSS-2 expression on pericytes was not determined. 

**Immunostaining**—Rabbit monoclonal ACE-2 primary antibody (Clone: SN0754, CAT# MA5-32307) and goat anti-rabbit Alexa Fluor-488 secondary antibody (CAT# A-11034) were purchased from Invitrogen. All staining steps were performed at RT. Astrocytes and pericytes were fixed with 4% paraformaldehyde for 10 min. and then incubated in a blocking buffer (10% goat serum in PBS) for 40 min. Cells were incubated with ACE-2 primary antibody (5 µg/mL) in 0.1% TWEEN-PBS or PBST (Phosphate Buffered Saline with Tween 20 (0.1%)) overnight at 4 °C. Following three PBST washes, cells were incubated with a secondary antibody at 1:50 dilution for 2-hrs at 37 °C. Then, the cells were washed with PBST three times and were air dried before mounting with DAPI solution (Prolong^TM^ diamond antifade mount CAT# P36962, Thermofisher Scientific, Waltham, MA, USA). Sample images were obtained using a Zeiss confocal microscope at the University of Nebraska Medical Center Advanced Microscope Core Facility, University of Nebraska, Omaha, NE, USA.

**RNA extraction and cDNA synthesis**—Total RNA was extracted from 8–10 × 10^6^ cells stored in RLT buffer (600 µL) according to the manufacturer’s instructions (Qiagen, Germantown, MD, USA). The quantity of RNA was assessed with a nanodrop (NanoDrop ONE^C^, Thermofisher Scientific, Waltham, MA, USA). RNA (2 µg) was used as a source to synthesize cDNA using Superscript IV VILO Master Mix (Invitrogen, Chicago, IL, USA # 11754-050) according to the manufacturer’s instructions. 

**ACE-2, TMPRSS-2, and Neuropilin-1 quantification by ddPCR**—TaqMan ddPCR assays were designed to amplify ACE-2, TMPRSS-2, and Neuropilin-1 targets using primer-probe pairs listed in Table 1 and were purchased from IDT. In the ddPCR assay, 50 ng cDNA was mixed with a primer-probe mixture, and 12.5 µL of 2× ddPCR super mix (Biorad, Hercules, CA, USA) was added in a total of 25 µL reaction mixture according to the manufacturer’s instruction. Samples were tested in four replicates (n=4) in the ddPCR assay in a 96 well plate (Biorad, Hercules, CA, USA # 12001925) and were sealed with Biorad Sealer PX1, and droplets were generated in AutoDG instrument (Biorad, Hercules, CA, USA). ddPCR assays were performed as described previously in our laboratory [37].

**SARS-CoV-2 infection of cells**—Pericytes, astrocytes, hBMVECs, and hLNECs were seeded at 0.25 million cells/well in a 12-well plate and following 24 hrs in an incubator (37 °C, 5% CO_2_). Cells were then infected with SARS-CoV-2 isolate USA-WI1/2020 (BEI; cat# NR-52384) at 1 MOI (multiplicity of infection). One-hour post viral infection, cells were washed with PBS to remove the cell-free virus and supplied with fresh medium. Cells and cell culture supernatants were collected at 24-, 48- and 72-h post-infection.

## 3. Statistics

ACE-2 mRNA expression was quantitated in astrocytes and pericytes from 3 and 4 donors respectively using three replicates. Subgroups were compared with a Mann–Whitney test. <0.05 *p* value was indicated by ‘*’, <0.01, ‘**’ and <0.001, ‘***’. 

## 4. Results

### 4.1. Moderate Levels of ACE-2 Was Expressed by Astrocytes and Pericytes

In astrocytes, 15.9 ± 1.3% (Mean ± SD, n = 2) cells expressed ACE-2 receptor, whereas, in pericytes, 23.1 ± 20.7% (Mean ± SD, n = 2) cells expressed ACE-2 but the expression was highly variable between the two donors (Figure 1A,B). In Vero cells, 18.6 ± 3.7% (Mean ± SD) cells expressed ACE-2 and served as a positive control. Interestingly, these ACE-2 positive cell numbers were at a saturated antibody concentration (6 µg/mL) as increasing antibody concentration from 2 to 6 µg/mL yielded similar results (Data not shown). Further evidence of ACE-2 expression on astrocytes and pericytes (Figure 1C) was obtained from immunocytochemistry experiments. Consistent with flowcytometry data, confocal images confirmed ACE-2 expression in both pericytes and astrocytes (Figure 1C).

ACE-2 mRNA copies were quantitated using specific Taqman probes in ddPCR assays. PBMCs, Jurkat cells, and Vero cells were used as controls. Mean mRNA copies of ACE-2 in astrocytes of three healthy donors was 95.53 ± 41.27 (Mean ± SD, n = 12) per 50 ng cDNA input. Interestingly, modestly but significantly lower (2.31 fold as compared with means, *p* value ≤.004) ACE-2 expression was observed in pericytes (42.1 ± 19.0, Mean ± SD, n = 9) compared with astrocytes (Figure 2A and Table 2). hPBMCs and Jurkat CD4 T cells showed a low level of ACE-2 mRNA expression. Whereas, very low levels of ACE2 mRNA copies were amplified in hBMVECs (2.4 ± 4.2, Mean ± SD, n=2) (Table 2 and Appendix A).

### 4.2. Pericytes Highly Express TMPRSS2 mRNA

The mean TMPRSS-2 copies expressed in astrocytes was 205.8 ± 126.7 (Mean ± SD, n = 12) from three donors and in pericytes was 667.2 ± 232.3 (Mean ± SD, n = 12) (Figure 2B and Table 2). Pericytes showed 3.24-fold significantly (*p* value ≤0.0001) higher TMPRSS-2 mRNA copies than astrocytes.

### 4.3. Astrocytes Highly Express Neuropilin-1, and Pericytes Show Moderate Expression 

Neuropilin-1 receptor expression was analyzed on both astrocytes and pericytes using flow cytometry. A higher percentage of astrocytes (56.4 ± 30.35%, Mean ± SD, n = 4) expressed Neuropilin-1 on their surface as compared with pericytes (30.35 ± 7.46%, Mean ± SD, n = 4) (Figure 3A,B). Neuropilin-1 expression was not detected on Vero cells confirming the brain cell specificity (Figure 3A). Neuropilin-1 mRNA was expressed in low copy numbers (1239 ± 1071, Mean ±SD, n = 3) in astrocytes as compared with pericytes (9307 ± 1852, Mean ± SD, n = 3) (Figure 4 and Table 2). 7.51-fold significantly (*p* value = <0.0001) higher Neuropilin-1 mRNA copies were detected in pericytes. In contrast to the surface expression of the Neuropilin-1 receptor on astrocytes, low mRNA copies were observed in ddPCR assays, and high mRNA copies were detected in pericytes.

### 4.4. Co-Expression of Entry Receptors on Astrocytes and Pericytes

Co-expression of ACE-2, TMPRSS2, and Neuropilin-1 was determined on astrocytes and pericytes extracted from a healthy donor. In astrocytes, 15%, 17.6%, and 95.6% of cells expressed ACE-2, TMPRSS2, and Neuropilin-1 respectively (Figure 5). Whereas, in pericytes, 37.8% and 82.9% of cells expressed ACE-2 and Neuropilin-1 respectively (Figure 5A). In Vero cells, 15%, 11.6%, and 0.26% cells expressed ACE-2, TMPRSS-2, and Neuropilin-1 respectively. Further analysis on ACE-2+ve astrocytes showed that 79.6% and 96.4% cells co-expressed TMPRSS-2 and Neuropilin-1 respectively and 42.5% ACE-2+ve cells co-expressed Neuropilin-1 (Figure 5B,C). In Vero cells, almost 100% of cells co-expressed TMPRSS-2 (Figure 5D).

### 4.5. SARS-CoV-2 Differentially Infects Astrocyte and Pericytes

To determine the infectivity of primary brain cells, astrocytes, and pericytes were infected with SARS-CoV-2 at 1 MOI. hBMVECs and hLNECs were not infected with SARS-CoV-2, which confirmed that endothelial cells were not susceptible to SAR-CoV-2 infection. Astrocytes (2.1 × 10^6^ ± 3.9 × 10^5^ (mean ± SD) genome equivalent copies/mL) and pericytes (5.3 × 10^6^ ± 1.0 × 10^5^ (mean ± SD) genome equivalent copies/mL) were infected with SARS-CoV-2 and infection peaked at 24 h (Figure 6). A moderate decline in viral load was observed at 72 h post-infection (1.45× 10^6^ ± 6.8 × 10^5^ (mean ± SD) genome equivalent copies/mL in astrocytes 3.9 × 10^5^ ± 5.7 × 10^4^ (mean ± SD) genome equivalent copies/mL in pericytes, Figure 6). In astrocytes and pericytes, we observed a >2-fold increase in viral RNA copies at the peak of infection as compared with baseline infection. 

## 5. Discussion

The contribution of ACE-2, TMPRSS-2, and Neuropilin-1 receptors present on astrocytes and pericytes for SARS-CoV-2 infection is unknown. In this present study, we analyzed the expression of ACE-2, TEMPRSS-2, and Neuropilin-1 on the surface of human brain-derived pericytes and astrocytes at the mRNA and protein level by employing gene expression, flow-cytometry, and imaging methodologies. The low amount of variable ACE-2 expression (23.1 ± 20.7%, Mean ± SD) observed on pericytes, is in agreement with an earlier report on mouse pericytes [38] including the variable expression observed in some healthy donors. This was further confirmed by Immunostaining (Figure 1C). mRNA level ACE-2 expression using specific Taqman probes precisely quantitated the ACE-2 mRNA copies in ddPCR assays. ACE-2 mRNA copies were 2.3-fold (*p* value ≤ 0.004) higher in astrocytes compared to pericytes. Our multi-faceted approach provided substantial evidence of the expression of ACE-2 on brain astrocytes and pericytes suggesting possible involvement in SARS-CoV-2 infection.

Pericytes expressed a 3.4-fold higher number of TMPRSS-2 mRNA copies (Figure 2B) compared to astrocytes. TMPRSS-2 mRNA expression in human brain cells (hBMVECs) mirrors the finding by Torices et al. [39]. Thus, our results not only affirm the earlier study findings but provide additional data on the mRNA level expression of TEMPRSS-2 in brain cells. Next, Davies et al. showed Neuropilin-1 expression in brain regions, including its presence in astrocytes. However, Neuropilin-1 mRNA and protein expression in pericytes at a cellular level remains unknown [36] and therefore, our results suggested the expression of Neuropilin-1 is present in pericytes. We observed high Neuropilin-1 mRNA copies in pericytes as compared with astrocytes. In contrast, lower Neuropilin-1 was expressed at the protein level. The reasons for these differences remain unclear, and future studies may focus on investigating these differences. Despite the differences in ddPCR and flow cytometry results, the data confirms the Neuropilin-1 expression in astrocytes and pericytes, suggesting the availability of a potential candidate receptor for SARS-CoV-2 on brain astrocytes and pericytes.

To understand the susceptibility and SARS-CoV-2 infection kinetics in primary astrocytes and pericytes, both cell types were infected with 1 MOI of SARS-CoV-2. Results indicated that pericytes were infected with SARS-CoV-2 but were still variable among different donors (Figure 5). All four-donor derived pericytes had similar infection kinetics reaching maximum viral RNA copies at 24 h post SARS-CoV-2 infection (Figure 6). The trends of infection in these cells were different from standard cell lines such as Vero, where infection usually peaks at 72 h time point then declines later time points. However, based on SARS-CoV-2 viral RNA copies data, astrocytes were infected, and viral RNA copies are higher than pericytes following infection with the same amount of input virus (Figure 6).

In conclusion, this report provides evidence of the presence of ACE-2, TMPRSS-2, and Neuropilin-1 proteins on primary pericytes and astrocytes and has demonstrated infection of SARS-CoV-2 in these cells. However, detailed time course studies are still needed to fully understand whether SARS-CoV-2 fully replicates in these cells. One limitation of our study is the lack of complementation or knockout of ACE-2, TMPRSS2, and Neuropilin-1 expression either at the mRNA level or blocking their expression on their surface. These studies are critical and may provide direct involvement of each of these receptors in SARS-CoV-2 infection in astrocytes and pericytes. Our data show that ACE-2, TMPRSS-2, and Neuropilin-1 expression on brain cells and potential contribution to SARS-CoV-2 infection. Considering the neurological symptoms and long COVID-19-related CNS complications observed following some COVID-19 infections, identifying the cells susceptible to SARS-CoV-2 infection and the contributing cellular receptors or factors that increase their susceptibility to SARS-CoV-2 infection has become highly important. Our study findings provide insight into the neurotropism of SARS-CoV-2. “In vivo” brain cell infection kinetics is still under investigation, and our future studies will focus on investigating these details and developing strategies to counter the effects of SARS-CoV-2 infection entry, blocking productive infection of pericytes to prevent passage through the BBB. However, the findings of the present study are helpful to provide a primary platform for future studies targeting novel strategies to inhibit SARS-CoV-2 entry into brain tissues.

## Figures and Tables

**Figure 1 ijms-24-08622-f001:**
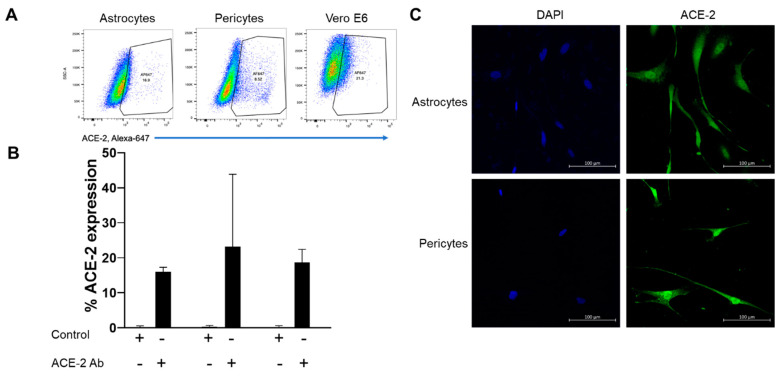
(**A**) Expression of ACE-2 receptor on human brain Astrocytes and Pericytes as analyzed by Flow cytometry; (**B**) Percentage of Astrocytes, Pericytes and Vero E6 cells expressing ACE-2 receptor and (**C**) “in situ immunostaining” of Astrocytes and Pericytes. Blue color represents DAPI bound to nuclei and the green color represents cell surface ACE-2 expression.

**Figure 2 ijms-24-08622-f002:**
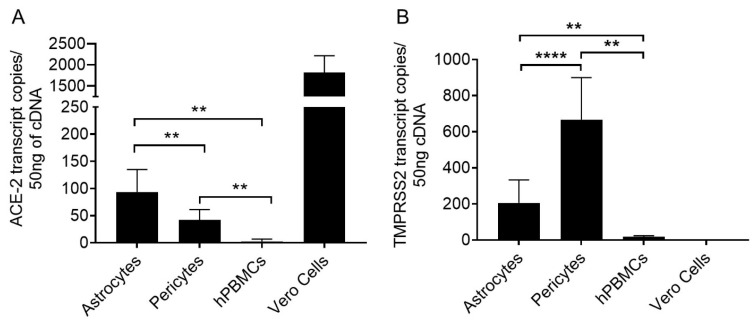
ACE-2 mRNA copies quantitated in human brain astrocytes and pericytes quantitated in ddPCR assays. Specific primer/probe sets are used to amplify (**A**) ACE-2, (**B**) TMPRSS2 in astrocytes and pericytes derived from three independent donors with three replicates are shown. *p* < 0.01 (**) and *p* < 0.0001 (****).

**Figure 3 ijms-24-08622-f003:**
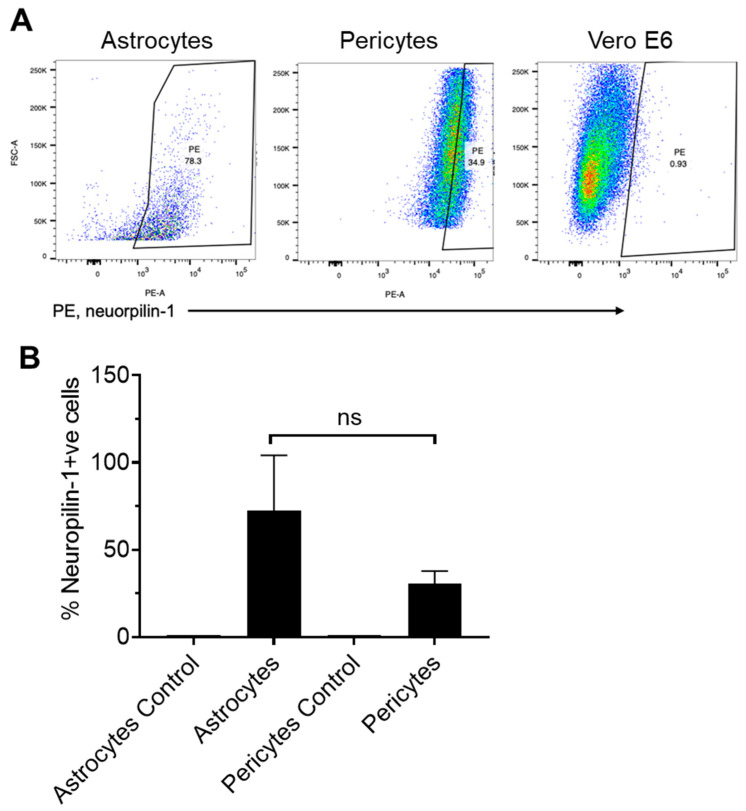
(**A**) Neuropilin-1 expression on human brain astrocytes and pericytes as analyzed by Flow cytometry and (**B**) Percentage of Neuropilin-1 positive cells. Subgroups were compared with Mann Whitney ‘T’ test. >0.05 *p* value was considered as non-significant (ns).

**Figure 4 ijms-24-08622-f004:**
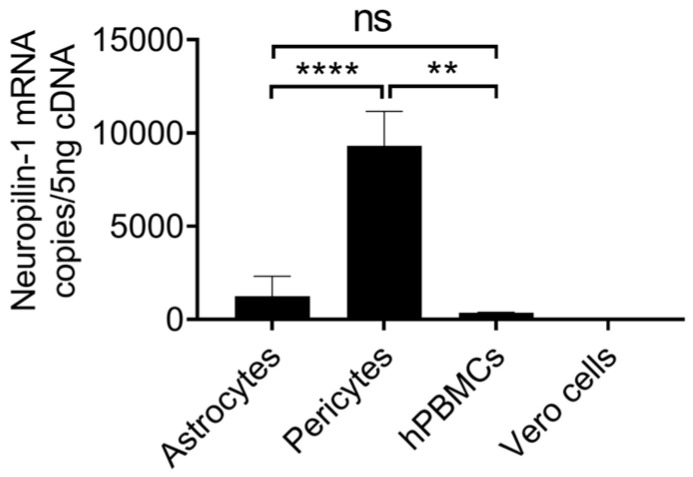
Neuropilin-1 mRNA copies quantitated in Astrocytes and Pericytes using ddPCR assays. Human PBMCs, and Vero cells are used as controls. Data from three independent donors using four replicates are shown. Subgroups were compared with the Mann–Whitney test. <0.05 *p* value was indicated by ‘*’; <0.01 *p*, ‘**’and <0.0001 *p*, ‘****’.

**Figure 5 ijms-24-08622-f005:**
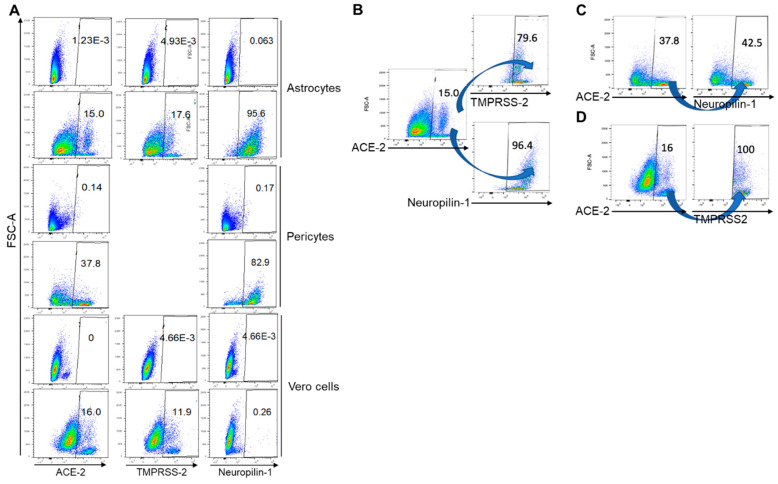
(**A**) Expression of ACE-2, TMPRSS2, and Neuropilin-1 expression on Astrocytes; ACE-2 and Neuropilin-1 on Perocytes and ACE-2, TMPRSS2, and Neuropilin-1 on Vero cells; (**B**) TMPRSS-2 and Neuropilin-1 co-expressing cells in ACE-2 positive astrocytes; (**C**) Neuropilin-1 co-expressing cells in ACE-2 positive pericytes and (**D**) TMPRSS-2 co-expressing cells in ACE-2 positive Vero cells.

**Figure 6 ijms-24-08622-f006:**
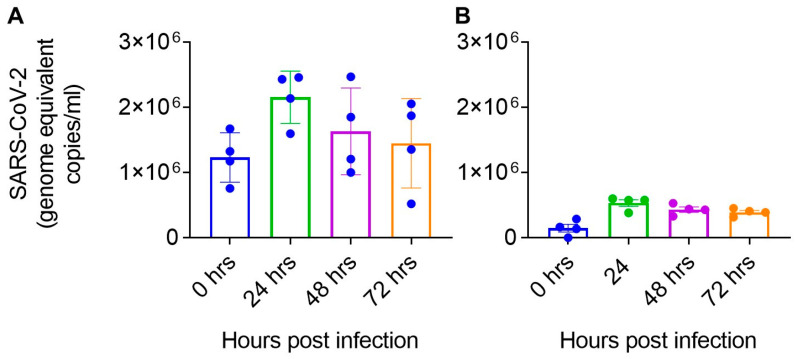
SARS-CoV-2 RNA copies in (**A**) pericytes and (**B**) astrocytes following viral infection. The four filled circles in each bar represent four independent donor primary cells used to infect 1 MOI of SARS-CoV-2.

**Table 1 ijms-24-08622-t001:** List of Primer-Probes used to amplify ACE-2, TMPRSS2, and Neuropilin target genes.

S. No.	Target Proteins	Primers	Sequences
1	ACE-2	Forward	5′-CCCATGATGAAACATACTGTGAC-3′
		Probe	5′-CCCGCATCTCTGTTCCATGTTTCT-3′
		Reverse	5′-TGGTAAAGGGTCCTTGTGTAAT-3′
2	TEMPRESS2	Forward	5′-CCTAGTGAAACCAGTGTGTCTG-3′
		Probe	5′-CATGATGCTGCAGCCAGAACAGC-3′
		Reverse	5′-CACCCGGAAATCCAGCAG-3′
3	Neuropilin-1	Forward	5′-ACACACACCAAAGCCAATTTC-3′
		Probe	5′-CTCCAACGGGTCCAGAAACAAGCC-3′
		Reverse	5′-TCTGTCTCCCGCTCATCTT-3′

**Table 2 ijms-24-08622-t002:** Summary of Mean mRNA copies ± SD per using 50 ng cDNA extracted from the total RNA from astrocytes, pericytes, hBMVECs, CD4 T cells and Vero cells.

Receptor	Astrocytes	Pericytes	hBMVECs	CD4 T cells	Vero cells
ACE-2	95.5 ± 41.2	41.2 ± 16.3	2.4 ± 4.2	30.2 ± 4.9	1925 ± 331.5
TEMPRSS2	205.8 ± 126.7	667.2 ± 232.3	17.6 ± 6	59.7 ± 11.7	0.4 ± 0.8
Neuropilin-1	1239 ± 1071	9307 ± 1852	355 ± 33	10.4 ± 1.4	0.0 ± 0.0

## Data Availability

Data sharing is not applicable to this article.

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
