# Peer review of "ACE-2, TMPRSS2, and Neuropilin-1 Receptor Expression on Human Brain Astrocytes and Pericytes and SARS-CoV-2 Infection Kinetics"

_ijms, 2023, doi:10.3390/ijms24108622_

Round 1

Reviewer 1 Report

The paper compared the levels of Ace2, TMPRSS2, and neurophilin-1 in pericytes, microvascular endothelial cells and astrocytes extracted from human brain. I found this manuscript rather light on useful experiments and on thoroughness in providing meaning full insight into these proteins expression level.

1. Ace2 expression level is examined by flow cytometry, mRNA levels, and immunofluorescence while there is only mRNA levels for TMPRSS2 (no flow or immunofluorescence) and neurophilin-1 is lacking the imunofluorescnece.

2. Infection assays appear rather low.  A) why is there only a 2-fold increase from 0 hours post infection to 24 hours post infection?) B) why does the infection decrease over time? C) what fraction of the cells are actually being infected?  Are these SARS-CoV-2 RNA levels virus that is being produced by the cell? Are they virus bound by the cell that never infect get degraded ect? 

3) How are the expressions of Ace2, TMPRSS2, and neurophilin-1 relative to each other?  Do cells expressing Ace2 also express TMPRSS2? or Neurophilin-1? and other such combinations?  Cells with Ace2 but no TMPRSS2 - what proteases are present that allow for spike cleavage and entry (ie. what are the Cathepsin B/L levels)?  

4) Starting with your opening sentence of your abstract you miss describe TMPRSS2 as a receptor. It is a protease that activates by cleavage of the S2' site after binding to the host cell receptor Ace2 (Bestle et al Life Sci Alliance 2020)  Cells expressing TMPRSS2 but not Ace2 will not infect with SARS-CoV-2. But cells expressing Ace2 will infect using difference proteases such as CathepsinB/L.

Also, TMPRSS2 mediate cleavage during virus entry requires the virus to be cleaved by furin at the S1/S2 site during viral production (Kreutzberger et al PNAS 2022, Essalmani et al JVI 2022). In the absence of furin process virus can still enter the cell through the cathepsin dependent cleavage pathway.  I think this manuscript would be superbly enhanced if the levels of furin, cathepsin B/L would also be examined - especially all the receptors and proteases relative to each other in each cell type. 

5) Can the expression of receptors relative to each other and proteases relative to each other be map in a cell infection assay in cells that are successfully infected with SARS-CoV-2 to understand if there are receptor or protease preferences for successful infection. 

Author Response

Point-by-Point Response to Reviewers Comments

Dear Editor,

     We thank you and the reviewer-1 for the opportunity to revise our manuscript entitled, “ACE-2, TMPRSS2, and Neuropilin-1 Receptor Expression on Human Brain Astrocytes and Pericytes and SARS-CoV-2 Infection Kinetics” and to improve the quality of our work. Here we provided point-by-point response to all reviewer-1’s comments. Hope our response to the reviewer-1 comments and the revised manuscript with suggested changes will be helpful to the reviewer to take a positive decision on the manuscript to recommend it for publication in the International Journal of Molecular Sciences.

Response to the Reviewer-1’s comments:

The manuscript compared the levels of ACE-2, TMPRSS-2, and neurophilin-1 in pericytes, microvascular endothelial cells and astrocytes extracted from human brain. I found this manuscript rather light on useful experiments and on thoroughness in providing meaning full insight into these proteins expression level.

Thank you for your opinion and comments on the manuscript. Please find our response to the comments below.

  1. ACE-2 expression level was examined by flow cytometry, mRNA levels, and immunofluorescence while there is only mRNA levels for TMPRSS2 (no flow or immunofluorescence) and neurophilin-1 is lacking immunofluorescence.

Res:  As the reviewer has requested, in a healthy donor brain derived astrocytes, flow-cytometry data was generated on co-expression of ACE-2, TMPRSS2 and Neuropilin-1 and ACE-2 and Neuropilin-1 expression on pericytes. In the revised manuscript, this data can be found in Figure 5 and can also made available below for the reviewer’s convenience.

Figure 5. A) Expression of ACE-2, TMPRSS2 and Neuropilin-1 expression on Astrocytes; ACE-2 and Neuropilin-1 on Perocytes and ACE-2, TMPRSS2 and Neuropilin-1 on Vero cells; B) TMPRSS-2 and Neuropilin-1 co-expressing cells in ACE-2 positive astrocytes; C) Neuropilin-1 co-expressing cells in ACE-2 positive pericytes and D) TMPRSS-2 co-expressing cells in ACE-2 positive Vero cells.

  1. Infection assays appear rather low.  A) why is there only a 2-fold increase from 0 hours post infection to 24 hours post infection?) B) why does the infection decrease over time? C) what fraction of the cells are actually being infected?  Are these SARS-CoV-2 RNA levels virus that is being produced by the cell? Are they virus bound by the cell that never infect get degraded ect? 

Res: A) We observed high number of viral copies at 0h post infection. Whether virus observed at 0h is due to some level of virus stuck to surface of cells or other possibilities, currently unknown and we could not be able to investigate further in this study due to technical challenges. A  two-fold increase in viral replication from 0h to 24h is possibly due to low level of non-productive viral replication.  

  1. B) The reasons for reduction of viral RNA copies following day 1 peak are unknown. Future studies are needed to show the reasons for this viral kinetic profiles.
  2. C) We did not measure the percent of cells infected in this experiment. SARS-CoV-2 RNA copies were determined in culture supernatants following isolation of cells and represent the viral RNA copies produced by cells. However, infectious nature of these viruses remains to be investigated.

3) How are the expressions of Ace2, TMPRSS2, and neurophilin-1 relative to each other?  Do cells expressing Ace2 also express TMPRSS2? or Neurophilin-1? and other such combinations?  Cells with Ace2 but no TMPRSS2 - what proteases are present that allow for spike cleavage and entry (ie. what are the Cathepsin B/L levels)?  

Res: We thank the reviewer for this important question. We evaluated the ACE-2. TMPRSS2 and Neuropilin expression in one healthy donor brain extracted astrocytes and ACE-2 and Neuropilin expression on pericytes. However, due to some technical challenges, we could not generate TMPRSS2 data in pericytes. This information is found in Figure 5 in the revised manuscript.

4) Starting with your opening sentence of your abstract you miss describe TMPRSS2 as a receptor. It is a protease that activates by cleavage of the S2' site after binding to the host cell receptor Ace2 (Bestle et al Life Sci Alliance 2020)  Cells expressing TMPRSS2 but not Ace2 will not infect with SARS-CoV-2. But cells expressing Ace2 will infect using difference proteases such as CathepsinB/L. Also, TMPRSS2 mediate cleavage during virus entry requires the virus to be cleaved by furin at the S1/S2 site during viral production (Kreutzberger et al PNAS 2022, Essalmani et al JVI 2022). In the absence of furin process virus can still enter the cell through the cathepsin dependent cleavage pathway.  I think this manuscript would be superbly enhanced if the levels of furin, cathepsin B/L would also be examined - especially all the receptors and proteases relative to each other in each cell type.

Res: As noted above, we provided ACE-2, TMPRSS2 and Neuropilin-1 co-expression on astrocytes using a antibody cocktail and show that astrocytes expressed 15% ACE-2, 17.6% TMPRSS2 and 95.6% neuropilin-1 expression. We apologize for not providing important information on TMPRSS2 expression in pericytes due to some technical difficulties, furin and Cathepsin B/L protein expression on astrocytes and pericytes due time constraints and is a subject of future studies. But will continue to work on other receptors and will provide this information in a future study.

5) Can the expression of receptors relative to each other and proteases relative to each other be map in a cell infection assay in cells that are successfully infected with SARS-CoV-2 to understand if there are receptor or protease preferences for successful infection. 

Res: SARS-CoV2 infection experiments in BSL3 labs were conducted overcoming several difficulties. In addition, these infection experiments with primary astrocytes and pericytes invited further difficulties with availability of low number of primary cells obtained from the same donor to match the SARS-CoV2 expression data with infection data. So, in this work, we could not be able to provide this useful information and is a subject of future study.

Reviewer 2 Report

Although the manuscript is quite interesting, it presents many flaws that must be resolved. In particular:

Lines 44-46: The multifaceted role of SARS-CoV-2 infection deserves to be highlighted. In particular, it deserves to be pointed out that SARS-CoV-2 infection can also lead to non-respiratory diseases such as Preeclampsia and male infertility (as recently reviewed PMID: 35114008, 35943095). This is an important point to add since authors investigated the role of this infection in brain, a non-respiratory organ. 

Line 70: ACE-2 and TMPRSS2 must be written in full length since are mentioned for the first time  

Line 80 and 83: what BMVECs and BMECs are?  a definition of these cell lines must be added

Line 82: correct ACE2 with ACE-2. 

Figure 1C: scale bars are needed

Line 112: in p<.0001 an asterisk is missed since there are three asterisks as well as p<.001. What about the two asterisks? 

Line 97: this sentence "(Add the control flow graph as an extra row)" is left from the authors revision and must be deleted 

Line 116: this sentence "Where is the flow data of TMPRSS2?" is left from the authors revision and must be deleted 

Figure 3A: Correct neuorpilin-1 with neuropilin-1

Line 142: what hLECs are?

Line 235: FBS and P/S must be written in full length

An accurate revision of typing errors is recommended 

Author Response

Dear Editor,

     We thank you and the reviewer-2 for the opportunity to revise our manuscript entitled, “ACE-2, TMPRSS2, and Neuropilin-1 Receptor Expression on Human Brain Astrocytes and Pericytes and SARS-CoV-2 Infection Kinetics” and to improve the quality of our work. Here we provided point-by-point response to all reviewer-2’s comments. Hope our response to the reviewer-2 comments and the revised manuscript with suggested changes will be helpful to the reviewer to take a positive decision on the manuscript to recommend it for publication in the International Journal of Molecular Sciences.

Reviewer-2 comments:

Although the manuscript is quite interesting, it presents many flaws that must be resolved. In particular:

Lines 44-46: The multifaceted role of SARS-CoV-2 infection deserves to be highlighted. In particular, it deserves to be pointed out that SARS-CoV-2 infection can also lead to non-respiratory diseases such as Preeclampsia and male infertility (as recently reviewed PMID: 35114008, 35943095). This is an important point to add since authors investigated the role of this infection in brain, a non-respiratory organ. 

Res: Authors thank the reviewer for his suggestion and multifaceted role is highlighted in the introduction part of the revised manuscript (Lines 79 to 82).

Line 70: ACE-2 and TMPRSS2 must be written in full length since are mentioned for the first time 

Res: ACE-2 and TMPRSS2 abbreviations are described when used for the first time in the revised manuscript

Line 80 and 83: what BMVECs and BMECs are?  a definition of these cell lines must be added

Res: Apologies for the confusion. Both human brain microvascular endothelial cells (BMVECs) and BMECs are the same but BMVECs are most frequently used abbreviation and is defined in the revised manuscript.

Line 82: correct ACE2 with ACE-2. 

Res: Corrected

Figure 1C: scale bars are needed

Res: Scale bars are included in the updated Figure 1C

Line 112: in p<.0001 an asterisk is missed since there are three asterisks as well as p<.001. What about the two asterisks? 

Res: The asterisks were included in the revised manuscript

Line 97: this sentence "(Add the control flow graph as an extra row)" is left from the authors revision and must be deleted 

Res: The revision statement was removed in the revised manuscript

Line 116: this sentence "Where is the flow data of TMPRSS2?" is left from the authors revision and must be deleted 

Res: The revision statement was removed in the revised manuscript

Figure 3A: Correct neuorpilin-1 with neuropilin-1

Res: Corrected

Line 142: what hLECs are?

Res: hLECs are human primary lymph node endothelial cells. To further clarify, hLECs are modified to hLNECs (human primary lymph node endothelial cells)

Line 235: FBS and P/S must be written in full length

Res: FBS and P/S abbreviations are described in the revised manuscript

An accurate revision of typing errors is recommended 

Res: The manuscript is thoroughly reviewed and typing errors are corrected.

Round 2

Reviewer 1 Report

This paper is suitable for publication and should be accepted in its current form. 

Reviewer 2 Report

the manuscript has been significantly improved and can be accepted in the present form